# Precision Medicine in Erythropoietin Deficiency and Treatment Resistance: A Novel Approach to Management of Anaemia in Chronic Kidney Disease

Nava Yugavathy [1], Bashar Mudhaffar Abdullah [2], Soo Kun Lim [3], Abdul Halim Bin Abdul Gafor [4], Muh Geot Wong [5,6], Sunita Bavanandan [7], Hin Seng Wong [8] and Hasniza Zaman Huri [1,*]

1   Department of Clinical Pharmacy and Pharmacy Practice, Faculty of Pharmacy, Universiti Malaya, Kuala Lumpur 50603, Malaysia; navayugavathy90@gmail.com
2   Clinical Laboratory Technology Department, Al-Rafidain University College, Baghdad 46036, Iraq; basharmudhafar22@ruc.edu.iq
3   Department of Medicine, Faculty of Medicine, Universiti Malaya, Kuala Lumpur 50603, Malaysia; limsk@ummc.edu.my
4   Faculty of Medicine, National University of Malaysia, Bangi 43600, Malaysia; halimgafor@gmail.com
5   Department of Renal Medicine, Royal North Shore Hospital, Sydney, NSW 2065, Australia; mwong@georgeinstitute.org.au
6   The George Institute for Global Health, University of New South Wales, Kensington, NSW 2052, Australia
7   Department of Nephrology, Hospital Kuala Lumpur, Kuala Lumpur 50586, Malaysia; sbavanandan@gmail.com
8   Department of Nephrology, Hospital Selayang, Batu Caves 68100, Malaysia; hinseng@gmail.com
*   Correspondence: hasnizazh@um.edu.my; Tel.: +60-3-79674909; Fax: +60-3-79674964

**Abstract:** The study of anaemia is a well-developed discipline where the concepts of precision medicine have, in part, been researched extensively. This review discusses the treatment of erythropoietin (EPO) deficiency anaemia and resistance in cases of chronic kidney disease (CKD). Traditionally, erythropoietin-stimulating agents (ESAs) and iron supplementation have been used to manage anaemia in cases of CKD. However, these treatments pose potential risks, including cardiovascular and thromboembolic events. Newer treatments have emerged to address these risks, such as slow-release and low-dosage intravenous iron, oral iron supplementation, and erythropoietin–iron combination therapy. Another novel approach is the use of hypoxia-inducible factor prolyl hydroxylase inhibitors (HIF-PHIs). This review highlights the need for precision medicine targeting the genetic components of EPO deficiency anaemia in CKD and discusses individual variability in genes such as the erythropoietin gene (EPO), the interleukin-β gene (IL-β), and the hypoxia-inducible factor gene (HIF). Pharmacogenetic testing aims to provide targeted therapies and interventions that are tailored to the specific characteristics of an individual, thus optimising treatment outcomes and minimising resistance and adverse effects. This article concludes by suggesting that receptor modification has the potential to revolutionise the treatment outcomes of patients with erythropoietin deficiency anaemia through the integration of the mentioned approach.

**Keywords:** precision medicine; erythropoietin deficiency anaemia; HIF-PHI; pharmacogenetics; inflammation

## 1. Introduction

Anaemia is a common complication of chronic kidney disease (CKD) and can significantly impact the quality of life of affected individuals. The cause of anaemia in patients with deteriorating kidney function is multifactorial. As the most common causes, when iron and nutrient deficiencies are ruled out, EPO deficiency is the most likely diagnosis. However, chronic inflammation, blood loss, and hyperparathyroidism are also taken into account. Treatment for anaemia among patients with CKD has focused on administering

erythropoietin-stimulating agents (ESAs) and iron supplementation. Methoxy polyethylene glycol-epoetin beta (Mircera) and Darbepoetin alfa (Aranesp) are two of the most commonly used ESAs. They function by increasing the production of red blood cells, which helps to alleviate anaemia symptoms. However, these drugs cause an undesirable number of adverse effects in a certain group of patients [1,2].

Recently, new treatments have offered additional options for the management of erythropoietin (EPO) deficiency anaemia in relation to CKD. One approach is modifying conventional iron therapy [3]. Intravenous iron, oral iron, and erythropoietin–iron combination therapy have all been used to improve iron reserves and support red blood cell production. However, oral ferrous ($Fe^{2+}$) compounds can cause gastrointestinal side effects, and intravenous iron (IV) therapy can be associated with the risk of anaphylaxis [4]. Therefore, newer formulations of IV iron administration with slow release and low dosages are being considered.

Another novel approach is the usage of hypoxia-inducible factor prolyl hydroxylase inhibitors (HIF-PHIs), roxadustat (Evrenzo), and daprodustat (Duvroq). HIF-PHIs increase endogenous erythropoietin production and red blood cell production, which can improve the symptoms of anaemia without posing the risks associated with exogenous erythropoietin. The OLYMPUS and DOLOMITES clinical trials have shown that roxadustat is effective in treating cases of anaemia with various comorbidities, presenting a favourable safety profile [5,6]. HIF-PH is also preferred in certain groups of CKD patients, including those suffering from heart failure [7] and chronic inflammation [8].

One of the greatest unmet needs in relation to EPO deficiency is the identification of causative genetic mutations, which could facilitate personalised medicine. A broad range of genes are involved in EPO deficiency. However, a molecular diagnosis needs to be established in a clinical setting to avoid unnecessary costly and invasive treatments. Of the many genes involved in this mechanism, we focus on the erythropoietin gene (EPO), the interleukin-β gene (IL-β), and the hypoxia-inducible factor gene (HIF).

This review article focuses on the three approaches to treating EPO deficiency anaemia in patients with CKD. We discuss current treatment options, practical advantages and shortcomings, precision molecular medicine targeting genetic components, and emerging alternative therapies.

## 2. Overview of Pharmacogenetics

Pharmacogenetics focuses on the influence of single-gene polymorphisms on drug responses. This process depends on the variations in how people respond to medicinal therapy, which is a rapidly expanding field in molecular biology and clinical medicine. With the advent of genetic testing, pharmacogenetics, a term that gained popularity in the 1930s, has now been rediscovered [9]. Up to 95% of the variability in treatment responses can be attributed to genetic variables. However, other elements, including age, gender, physiology, pathophysiology, culture, behaviour, and environment, may significantly impact the variations in these parameters. For example, if members of the same family with the same inherited condition respond differently to the same medical treatment, genetic factors are likely to play a role [10].

A single-nucleotide polymorphism (SNP), often known as a genetic variation, is a change in a nucleotide sequence that affects the pharmacokinetic and pharmacodynamic characteristics of medications [11]. Pharmacogenetic tests are designed to identify patients who respond or do not respond to treatment, show interactions with other drugs, experience side effects, or need their dosages adjusted [10]. Pharmacogenetics has become a vital subject due to the concept of individualised medication. In the field of medicine known as "personalised medicine", treatment choices are based on a patient's entire dataset, for which their individuality is taken into account. These data consist of genetic, environmental, and quality-of-life information.

### 3. Impact of Iron Deficiency and Treatment

Chronic kidney disease is largely considered an inflammatory disease that affects haematopoietic function and the corresponding pathway [12]. There are two types of iron deficiency (ID) that disrupt EPO production, namely absolute and relative [13]. Absolute iron deficiency occurs when there is a diminished level of iron in the body due to halted iron absorption or severe blood loss. An affected individual will typically present with decreased iron, ferritin, and transferrin saturation and an increased total iron binding capacity (TIBC). Relative iron deficiency, in contrast, is due to the inefficient use of stored iron, which occurs due to inflammation, genetic errors, or EPO deficiency [4]. ID is also corelated with cardiovascular disease prognosis, for which a patient's iron status is often screened [14]. Campodonico et al. [15] reported that heart failure patients with transferrin saturation (TSAT) < 20% and a ferritin level above 100 ug/L are likely to have the worst outcomes.

Iron supplementation therapy is typically administered in the form of iron-containing oral supplements such as ferric maltol, a non-salt-based form of iron. Ferric maltol has been shown to be more effective than other iron salts, presenting fewer side effects in a phase III trial. Evidence from the 1-year AEGIS-CKD [16] trial has shown that this compound is able to normalise and sustain Hb levels with a much lower rate of gastrointestinal disorder than that of ferrous sulphate [17]. Although this compound has been licensed in the United States (US) and Europe for the treatment of iron deficiency anaemia [18], its efficacy and safety profile are still under assessment with respect to CKD patients. Newer intravenous iron supplementation treatments such as ferric carboxymaltose (FCM) and ferric derisomaltose (FDM) have been formulated to improve safety and reduce hypersensitivity reactions. FCM and FDM are highly concentrated forms of iron that utilise a unique iron–carbohydrate complex and can be administered quickly in a single or a few IV doses [19,20]. However, it is important to note that the administration of IV iron to patients with CKD requires careful monitoring and close collaboration between the patient's nephrologist and haematologist. This is because patients with CKD are susceptible to iron overload, which may lead to undesired side effects [21]. Additionally, patients with advanced CKD may have difficulty excreting excess iron, which can lead to iron overload and further complications. In the case of functional iron deficiency, which is usually caused by the underutilisation of EPO due to hepcidin upregulation, IV iron administration may cause iron overload toxicity and enhance oxidative stress. In an attempt to compare the safety profiles of FCM, FDM, and iron sucrose in terms of hypersensitivity using a robust and reliable method, Pollock and Biggar [22] concluded that the risk of hypersensitivity regarding FDM is relatively lower compared to the other two compounds.

### 4. Erythropoietin

In a normal kidney, renal EPO-producing cells (REPs) are peritubular interstitial fibroblast-like cells and pericytes of the cortical–medullary region [23] that control EPO gene expression, primarily through the hypoxia-inducible factor (HIF) pathway [24]. EPO acts as the primary hormone regulator of erythrocyte production or erythropoiesis in the bone marrow. It is also synthesised by the liver [25] and the brain; however, the amount synthesised by these tissues alone is insufficient to maintain adequate erythropoiesis [26]. According to Nangaku et al. [27], EPO deficiency may be caused by constant inflammatory cell infiltration. At the cellular level, the production of EPO in REPs is regulated by a number of EPO-producing cells via an "on" or "off" mechanism, which changes explicitly in response to hypoxia and or anaemia. Under normal conditions, REPs possess an EPO-producing ability but do not produce EPO (OFF-REPs). In healthy individuals, when oxygen supplies decrease or oxygen demands rise, OFF-REPs begin to produce EPO through HIF-mediated EPO gene transcription (ON-REPs) [28]. In CKD patients, however, sustained inflammation in renal fibrosis becomes the major contributor to EPO deficiency, as the damaged REPs in fibrotic kidneys transform into myofibroblasts, causing the concomitant repression of their ability to produce EPO [28].

## 5. Pathophysiology of Inflammation and Linked Genetic Factors

The serum erythropoietin levels in anaemic patients without renal failure but with inflammation due to other factors were found to be lower compared to these levels in similarly anaemic patients without inflammation, demonstrating the relationship between inflammation and impaired erythropoiesis [29,30]. Signalling mediators of pro-inflammatory and pro-fibrotic pathways, such as GATA-2 (binding factor), nuclear factor kappa-light-chain-enhancer of activated B cells (NF-KB), and transforming growth factor beta (TGF-β)/Smad, often play a role in the direct and indirect suppression of the EPO-regulatory mechanism [30,31].

The GATA family transcription factor GATA-2 binds to the GATA motif in the vicinity of the transcriptional initiation site of the EPO promoter in the -30 region [32]. Some evidence has shown that the functional activation of the GATA-2 signalling pathway by pro-inflammatory cytokines leads to the impairment of EPO production. Observations from one study revealed that GATA-2 suppressed EPO gene expression in cell cultures. When the TATA element was transfected, EPO gene promoter activity rose 2.5-fold; however, pro-inflammatory cytokines decreased EPO gene promoter activity in cells transfected with pGATAwt relative to control cells [31]. This result suggests that the signal transduction pathways of these cytokines may be modulated by GATA-2, inducing lower EPO production under inflammatory conditions.

NFκB signalling is also responsible for the negative regulation of EPO gene transcription via competition with co-activators of hypoxia-induced EPO gene expression [33]. The activity of transcription factor NF-kB is closely related to the processes of activation, proliferation, and cell differentiation into myofibroblasts [28]. Souma et al. [28] revealed that the sustained activation of NFκB signals in renal erythropoietin-producing cells results in a phenotypic transition. These findings show that although the NFκB signal is important with regard to repressing erythropoietin production, it is also associated with transforming EPCs into myofibroblasts.

Another major signalling pathway involved in the pathogenesis of fibrosis with the concomitant loss of EPO is the TGF-β/Smad3 pathway [34]. Activated TGF-β functions via Smad-dependent and -independent signalling pathways, which are major pathways in many pathophysiological processes of kidney diseases [35]. It has been well documented that Smad3 is a strong downstream mediator of renal fibrosis in diabetic nephropathy [36], hypertensive nephropathy [37], obstructive kidney disease, and glomerulonephritis [38]. In contrast, Smad2 acts as a reno-protective agent by competing with Smad3 signalling through phosphorylation and nuclear translocation [37].

Elevated levels of pro-inflammatory cytokines and inflammation-related indicators characterise renal fibrosis in CKD and the loss of EPO production [39]. Gene transcription and cytokine release may be affected by cytokine gene polymorphisms, which could modulate the risk of renal fibrosis and EPO deficiency anaemia progression [40]. As shown in Figure 1, Yan and Xu summarised the pathophysiology of the inflammatory cytokines that cause anaemia and EPO resistance [10].

The interleukin-6 (IL-6) gene is found on chromosome 7p21 and consists of five exons and four introns. IL-6 has several polymorphisms in the following promoter regions: −174 G to C and −597 G to A. IL-6 is rapidly expressed in a highly transient manner during inflammation. Mutations in rs2228145 have been associated with renal disease due to inflammation, and they also cause renal fibrosis [41]. A previous study on dialysis patients conducted by Sharples et al. [42] showed the influence of an IL-6 (−174 G/C) polymorphism on ESAs' responsiveness. They observed that there was a significantly higher ESA requirement in the GG and GC ACE genotypes compared with the CC group, which remained an independent association.

Tumour necrosis factor (TNF) is a pro-inflammatory cytokine produced by immune cells and adipocytes. In CKD, there is increased production of TNF as a result of inflammation and oxidative stress. Elevated TNF levels can have multiple effects on the body, including the suppression of EPO production by the kidneys [12]. TNF-α inhibits the

production of EPO in the kidneys by interfering with the transcription of the EPO gene and by promoting the destruction of EPO-producing cells. This inhibitory effect of TNF on EPO production contributes to the development of renal anaemia in CKD patients [43].

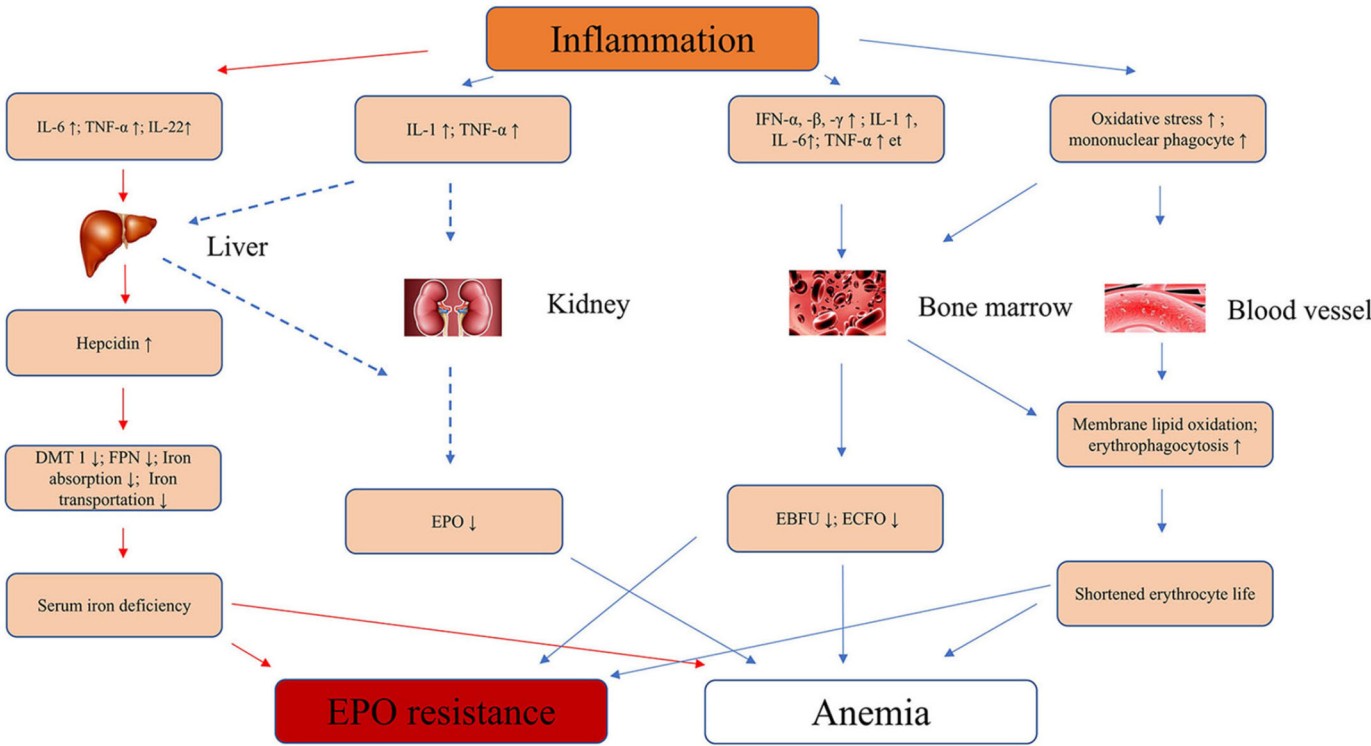

**Figure 1.** Inflammatory cytokines leading to anaemia. IL-6, interleukin-6; IL-1, interleukin-1; IFN-α, -β, -γ, interferon-α, -β, -γ; TNF-α, tumour necrosis factor-α; EBFU, erythroid burst-forming units; ECFO, erythroid colony-forming units; ↑, increase; ↓, decrease; FNP, ferroportin; DMT1, divalent metal transporter 1 ([12] (permission obtained)).

The IL-1 gene cluster on chromosome 2q contains three related genes within a 430-kb region, IL-1α, IL-1B, and IL-1RN, which encode the pro-inflammatory cytokines IL-1α and IL-1β. The IL-1β-511C/T (rs1143627) single-nucleotide polymorphism (SNP) has been associated with a variety of diseases in which inflammation plays an important role [44]. Jeong et al. [45] reported that the IL-1β-511CC genotype was significantly associated with lower ERI values among haemodialysis patients. In this study, patients with the IL-1β-511 TT genotype also showed a significantly higher mean IL-1β level (0.9 ± 0.6 pg/mL) compared to those with the IL-1β-511CC genotype (0.3 ± 0.3 pg/mL; $p = 0.02$) [45].

## 6. Erythropoietin Treatment and Resistance

According to the KDIGO guidelines (2013) [46], it is recommended that erythropoietin-stimulating agent (ESA) treatment should be considered for anaemic patients with CKD who are not on dialysis when their haemoglobin level is <10 g/dL. This treatment must be individualised based on the rate of decrease in haemoglobin levels, prior response to iron therapy, the likely requirement for a transfusion, the risks related to ESA therapy, and the presence of symptoms attributable to anaemia [47]. A third-generation ESA, darbepoetin alpha, which is modified from EPO via the insertion of a large pegylation chain (called the continuous EPO receptor activator) that increases the duration of the drug's effects, has been approved for sale on the market [48]. Although ESAs have been found to improve patients' quality of life, reduced haematopoietic responses to ESA treatment have been associated with an increased risk of adverse outcomes [49,50]. Patients with heart conditions are often associated with higher erythropoietin resistance [49,51]. Safety concerns regarding

ESA therapies have paved the way for the development of a new class of drugs. Table 1 lists the current treatment approaches [8].

**Table 1.** Summary of trials focusing on newer treatment approaches.

| Future Treatment Improvement Focus | Author/Year | Country of Study | Study Population, Sample Size (n) | Results | | |
|---|---|---|---|---|---|---|
| Clinical Trials | | | | | | |
| Comparison between weekly single dose and divided moderate dose of rHuEPO | Xiuling et al. [52] | China | Haemodialyzed (n = 88) | There was no significant difference in terms of medication safety in both groups | | |
| HIF-PH inhibition according to blood group | Funakoshi [53] | Japan | Haemodialyzed (n = 163) | Treatment Roxadustat Daprodustat | Blood group A O | Efficacy% 47 55 |
| Basic Experiments | | | | | | |
| Selective inhibition of PH2 | Su et al. [54] | China | In vivo (n = 25) | Ongoing trial | | |
| Ferroptosis as a potential therapeutic target in CKD | Wang et al. [55] | China | Mouse model (n = 24) | 1. Dysfunctional iron metabolism is an important contributor to ferroptosis. 2. Ferritinophagy was observed among CKD-afflicted rats. 3. Regulation of iron metabolism and TGF-β1/Smad3 pathway can interfere with the progression of renal damage. | | |

## 7. Precision Medicine in EPO Deficiency and Treatment Resistance

There has been a substantial amount of interest in creating a gene therapy approach with which to deliver EPO via a single infusion of the EPO gene and thus guarantee EPO delivery in the long term. In one attempt, investigators managed to establish a hypoxia-regulatory mechanism that was in line with the homeostatic system, thus characterising it as a natural approach. They also revealed that the promotor OBHRE region increased the haematocrit level gradually and safely in a relevant anaemic mouse model [56].

In a different approach, EPO from human cells (hEPO) was manufactured to develop a method whereby EPO could be directly secreted without any further formulation, to reduce the risk of antibody production. This study was carried out by Lippin et al. [57] using a Biopump transduced with clinical-grade Ad-MG/EPO-1 (Ad5 E1/E3 deleted) expressing hEPO, yielding a promising result.

However, it is important to note that gene therapy is still an emerging field and is not yet widely available for clinical use. Clinical trials are currently underway to evaluate the safety and efficacy of gene therapy with respect to various genetic conditions, including EPO deficiency caused by mutations in the EPO gene.

## 8. Genetic Factors of EPO Deficiency and Treatment Resistance

The human EPO gene is located on chromosome 7q21, which is well known for harbouring increased susceptibility to diabetic nephropathy [58]. Endogenous and recombinant EPO stimulates erythropoiesis by binding to the erythropoietin receptor (EpoR). Messenger RNA (mRNA) alternative splicing can give rise to the soluble form of the receptor (sEpoR), which lacks a transmembrane domain. sEpoR acts as an antagonist of EPO due to its higher affinity for EPO, which can lead to resistance towards ESA. Moreover, in vitro experiments conducted by Tong et al. [59] showed that a single-nucleotide polymorphism from G to T in the EPO promoter (rs1617640) could alter EPO mRNA levels [60]. A predisposition in this promoter region might also contribute to EPO deficiency in pre-dialysis patients and, therefore, needs to be investigated.

Pro-inflammatory genes are widely studied with respect to their role in anaemia associated with chronic diseases and are known to cause ESA resistance [61]. Jeong et al. [45]

reported that the IL-1β-511CC genotype was significantly associated with lower erythropoietin resistance values in haemodialysis patients. A polymorphism in this region was shown to reduce EPO mRNA expression and erythropoietin secretion in human hepatoma cell lines. According to Nangaku and Eckardt [27], this EPO deficiency may be due to constant inflammatory cell infiltration that leads to the decreased production of EPO. The roles of several pathways in inducing inflammatory conditions in renal fibrosis, which eventually leads to EPO deficiency anaemia, have been identified (Figure 2).

Hypoxia-inducible factor (HIF) was first discovered during the identification of a hypoxia-responsive element (HRE) in the erythropoietin gene in 1991 [62]. Besides regulating oxygen via signalling the EPO gene, it also plays a role in stem cell maintenance, growth factor signalling, epithelial–mesenchymal transition, angiogenesis, and metabolism [63,64]. Under normoxic conditions, HIF is rapidly ubiquitinated and degraded, a process primarily controlled by a family of oxygen-dependent prolyl hydroxylases (PH). However, hypoxia triggers the stabilisation of HIF-1α, which is then translocated from the cytoplasm to the nucleus and heterodimerises with HIF-1β, which protects it from Von-Hippel–Lindau (VHL)-mediated proteasomal degradation. The transcriptionally active HIF complex formed via heterodimerisation associates with HRE in the regulatory regions of target genes and binds to transcriptional coactivators to induce EPO gene expression [65,66].

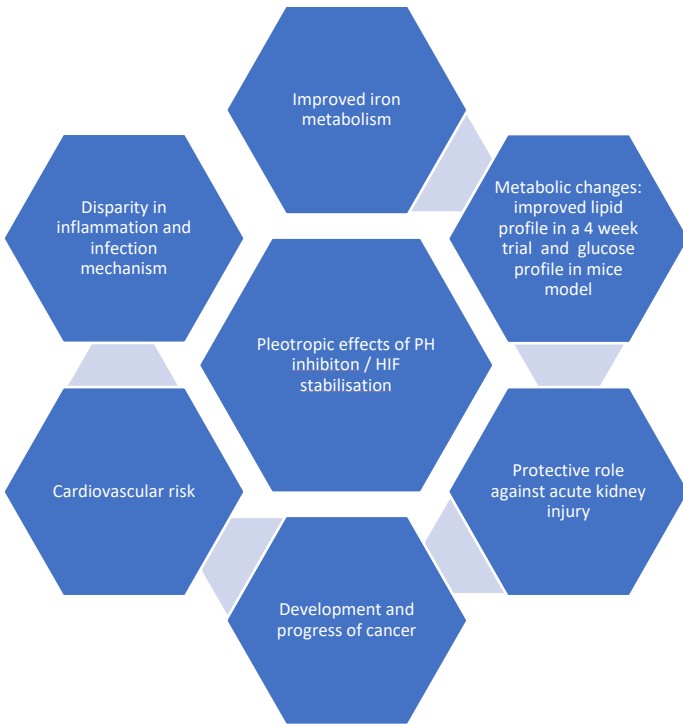

**Figure 2.** Pleiotropic effects of prolyl hydroxylase (PH) inhibition. HIF, hypoxia-inducible factor. Improved iron [67,68]; improved lipid profile [69]; glucose profile [52]; protective against kidney injury [70]; development of cancer [71], cardiovascular risk [72]; inflammation [73,74].

## 9. Hypoxia-Inducible Factor Prolyl Hydroxylase (HIF-PH) Inhibitors

HIF-PH inhibitors target the activity of prolyl hydroxylase (PH) enzymes, which play a key role in regulating the stability and activity of HIF [75]. PHs belong to the iron and α-ketoglutarate (α-KG)-dependent dioxygenase superfamily, and they have several identified isoforms, namely PH 1, 2, and 3 [76]. PHs exhibit overlapping but different tissue expression levels, and their subcellular localisation varies [67]. PH 2 is often associated with HIF-1α in normoxia regulation, and any loss of function relating to PH 2 causes familial erythrocytosis [68]. PH 3, on the other hand, is the regulator of HIF-2α and is found to be overexpressed in hypoxic conditions [71]. Several recent studies have investigated

the potential use of HIF-PH inhibitors as a treatment for CKD [77–79]. By blocking the activity of these enzymes, HIF-PH inhibitors can increase the activity of HIF and trigger cellular responses to low oxygen levels that regulate and stimulate erythrocyte production, potentially alleviating symptoms of anaemia.

CKD is associated with inflammation, which, in turn, causes functional iron deficiency and ESA hypo-responsiveness. HIF is believed to improve these two factors since it plays a crucial role in the regulation of cellular responses to low oxygen levels and reportedly reduces serum hepcidin levels [79]. Moreover, HIF-PH inhibitors were reported to decrease fibroblast growth factor (FGF)-23 levels in an animal model of CKD [80]. It was also noted by researchers that HIF-1$\alpha$ and its isoform 2$\alpha$ play a role in clear cell renal carcinoma (ccRC), but their functional significance is not clear in the developmental stage [81]. An earlier investigation revealed that tumour cells often present the overexpression of HIF-1$\alpha$. The overexpression of either isoform was also associated with cardiomyopathy [72]. However, recent evidence shows that these two isoforms have opposite outcomes [82]. The pleiotropic effects (Figure 2) of HIF have also led to the investigation of metabolic changes; for instance, an improvement in glucose tolerance and insulin sensitivity in a mouse model was observed by Sugahara et al. [83]. Both isoforms of HIF have been shown to be expressed in innate immunity [73], while HIF-1$\alpha$ has been revealed to be expressed in adaptive immunity [74].

In clinical trials, roxadustat (FG-4592) users experienced higher rates of upper respiratory infections, pneumonia, and urinary tract infections than the control group, which may have been related to the immunological regulation brought about by PH inhibition [5]. While it is true that HIF-PHIs have been well tolerated in clinical trials, there may still be some concerns regarding copper toxicity. Nakamura et al. [84] described four cases of individuals with renal anaemia who had elevated serum copper levels. Research on treatment with HIF-PHI medication, whose administration regimen was returned to normal after switching from HIF-PHI to darbepoetin alfa, indicated that HIF-PHI may be linked to elevated serum copper levels. Since a genome-wide analysis of the HIF transcriptome has shown that at least 500–1000 genes are under the control of HIF, stabilising this factor to improve the oxygen-sensing mechanisms might affect other complex pathophysiological characteristics in the long term.

In phase 3 trials, namely OLYMPUS and ROCKIES, Fishbane [5] reported that, regardless of inflammation, roxadustat improved haemoglobin levels in comparison to a placebo and an epotin alfa group. It was also proven to promote intestinal iron absorption and hepcidin reduction [85]. However, extensive clinical trials are required to study the long-term safety profile of roxadustat with respect to CKD inflammatory anaemia.

Daprodustat (GSK-1278863) is another potent HIF-PH 1–3 inhibitor, and it was first approved in Japan for renal anaemia treatment in June 2020 (GlaxoSmithKline) [86]. However, this molecule is still under investigation in many countries (NCT03029247; 205665; ASCEND-BP and NCT03457701; 201771; ASCEND-Fe); the FDA, after some hesitation, finally approved it in February 2023. Several other HIF-PH inhibitors are being developed and studied in order to target several patient cohorts—for example, dialysed and dependent patients.

Vadadustat (AKB-6548), developed by Akebia Therapeutics, inhibits all three PHs, with increased affinity towards PHD3 [75]. Vadadustat (Vafseo®) received marketing authorisation from the European Commission and the United Kingdom Medicines and Healthcare Products Regulatory Agency in April 2023 and May 2023, respectively. It is currently being used in 33 nations, including Japan. In the PRO$_2$TECT trial (NCT02648347 and NCT02680574), this drug did not meet its pre-specified safety endpoint in a non-dialysed patient cohort [87]. However, it showed good efficacy in terms of increasing haemoglobin levels, reducing serum ferritin and hepcidin levels, and improving TIBC [88].

Molidustat (BAY 85-3934) and enarodustat (JTZ-951) are also being tested in phase III clinical trials by different pharmaceutical companies, and they have presented considerable efficacy and safety profiles in phase 2 trials. Several review articles have summarised

the phase 3 clinical trials evaluating the efficacy of HIF stabilisers for the non-dialysis population (pertinent tables can be found in [89,90]). In the SYMPHONY ND trial, no adverse events occurred in the enarodustat arm; embolic and thrombotic events were previously observed in a Chuvash polycythemia patient. Retinal disorders were somewhat frequent. A limitation of this trial was its short duration, i.e., 24 weeks [91]. Although increased serum vascular endothelial growth factor (VEGF) levels were observed in the molidustat (Varenzin-CA1) group, the enarodustat (ENAROY)-treated patient did not exhibit the same trend [92].

## 10. Future Treatment Approaches

The newest molecules currently being researched are known as Tetrahydropyridin-4-ylpicolinoylglycines, which are intended to inhibit PH-2 [54]. This targeted approach might improve the selective binding and reduce some of the undesirable pleiotropic effects of HIF stabilisers. In the context of personalised medicine, an investigation of the association between an ABO blood group and HIF PH showed that roxadustat had better therapeutic efficacy for the A blood group, while daprodustat was more effective for the O blood group, with regard to haemodialyzed individuals [53]. While recombinant EPO treatment is not entirely disadvantageous for all patients, a more personalised approach to determining the treatment target would be beneficial. For instance, according to Joksimovic Jovic et al. [93], elevated serum ferritin levels and increased catalase activity contribute to ephemeral EPO resistance. Oxidative stress, poor nutrition, chronic inflammation, and vitamin D deficiency increase the risk of long-acting EPO resistance. Kidney cells are prone to iron overload, which causes ferroptosis, i.e., regulated cell death due to iron overload. This aggravates CKD and renal anaemia, leading to irregular iron metabolism [94]. It is believed that anaemia in CKD develops as the number of renal EPO-producing cells and the production of fibroblast-derived erythropoietin are decreased despite the tissue hypoxia caused by anaemia. As a result, EPO deficiency anaemia becomes the major cause of anaemia in CKD [23]. Wang et al. [55] attempted to demonstrate that ferroptosis could be a novel target approach in delaying CKD that eventually leads to renal anaemia.

## 11. Potential Approach from the Perspective of Drug Design and Development to Addressing Treatment Resistance in Erythropoietin Deficiency Anaemia: Receptor Modification

Receptor modification can occur through a variety of mechanisms, such as changes in a receptor's gene expression, post-translational modifications (such as phosphorylation, glycosylation, or acetylation), or alterations in a receptor's structure or conformation. These modifications can affect a receptor's sensitivity or specificity to certain ligands (molecules that bind to a receptor), its signalling efficiency or duration, or even its ability to interact with other proteins or downstream effectors. For example, the EpoR can be engineered to have a higher affinity for EPO, allowing it to bind to EPO more effectively and stimulate erythropoiesis more efficiently. Alternatively, the EpoR can be modified to have a longer half-life, which would prolong its activity and allow it to stimulate erythropoiesis for a longer period. Computational techniques are being used to model the erythropoietin receptor and perform molecular simulations. These simulations could help to identify potential binding sites, predict conformational changes, and assess the impacts of specific modifications on receptor function [95].

## 12. Conclusions

EPO deficiency has a complex mechanism that involves many pathways. Comorbidities such as dyslipidaemia, hypertension, and diabetes are also contributing factors in this condition. In addition, concomitant drugs, such as ACE inhibitors, statins, diuretics, anti-platelets, oral diabetic agents, and so on, are also widely investigated with respect to their roles as potential biomarkers. Identified or known pathways are being extensively researched to provide more personalised treatments and achieve good treatment efficiency. For example, drugs that target specific molecular pathways involved in EPO production

or responses are being investigated. These targeted therapies may be identified through genetic testing and personalised treatment plans. The investigation of biomarkers is also on the rise, and genetic contributions to pathophysiology should also be investigated. Some genetic polymorphisms and expression patterns may result in variations in individual responses to ESA treatment. Administering EPO based on the physiological demand could also reduce adverse events and unnecessary costs for patients.

**Author Contributions:** Conceptualisation, N.Y., B.M.A. and H.Z.H.; writing—original draft preparation, N.Y.; writing—review and editing, B.M.A., H.Z.H., S.K.L., A.H.B.A.G., M.G.W., S.B. and H.S.W.; supervision, H.Z.H. and S.K.L.; project administration, H.Z.H. and S.K.L.; funding acquisition, H.Z.H., S.K.L., A.H.B.A.G., M.G.W., S.B. and H.S.W. All authors have read and agreed to the published version of the manuscript.

**Funding:** This research was funded by the Fundamental Research Grant Scheme (FRGS), FP020-2016.

**Conflicts of Interest:** The authors have no conflict of interest to declare.

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
