# Peer review of "Precision Medicine in Erythropoietin Deficiency and Treatment Resistance: A Novel Approach to Management of Anaemia in Chronic Kidney Disease"

_cimb, doi:10.3390/cimb45080413_

Round 1

Reviewer 1 Report

Comments and Suggestions for Authors

In this manuscript, authors reviewed the mechanisms of erythropoietin-resistant anemia and recent approaches for managing renal anemia. The subject of the review seems to be interesting for many readers. However, there are some concerns in this manuscript. The reviewer’s comments are described as follows.

1. In Introduction section, authors emphasized cardiovascular and thrombotic risk of ESA as well as favorable safety profile of HIF-PHI. According to large clinical trials, however, cardiovascular and thrombotic risk of HIF-PHI appeared to be no less than that of ESA. There are obvious concerns about cardiovascular and thrombotic events with long-term use of HIF-PHI. Authors’ descriptions in the text can cause misunderstanding.  

2. In Iron Deficiency, recent reports suggest the important roles of increased transferrin in cardiovascular events. In addition, authors should show what levels of plasma iron can cause overload toxicity.

3. In Table 1, authors must not use the original table published in other paper, without permission from the publisher. Regarding all figures and tables in this manuscript, authors must describe the name of publisher and that they got permission from the publisher in the legends, if they use the same figures and tables.

4. In HIF-PHI section, authors described that only roxadustat and daprodustat are clinically available now. However, vadadustat, molidustat, and enarodustat are already available in some countries including Japan. Much data from large clinical trials using these five HIF-PHI agents should be available. More detail explanations about efficacy and safety are needed.

5. In Table 2, basic experiments and clinical data should not be combined.

Author Response

Please refer to attachment

Please find the changes/amendments in yellow highlight in the re submitted manuscript. 

Reviewer 2 Report

Comments and Suggestions for Authors

Review of the manuscript

cimb-2477528-peer-review

Precision Medicine in Erythropoietin Deficiency and Treatment Resistance: A Novel Approach in management of Anemia  of Chronic Kidney Disease

By Nava Yugavathy et al.

The manuscript is a review paper. The paper describes the current possibilities and future prospects for the treatment of anemia developing in the course of chronic kidney disease (CKD). In the discussion, the authors focus primarily on genetic therapy, erythropoietin analogues or agents that inhibit the degradation of HIF-1, which is responsible for the stimulation of erythropoietin secretion in the course of CKD.

The reviewed paper refers to an article previously published by the same authors: Yugavathy N, Huri HZ, Kun LS, Bin Abdul Gafor AH, Geot WM, Bavanandan S, Seng WH. Clinical and genetic markers of erythropoietin deficiency anemia in chronic kidney disease (predialysis) patients. Biomark Med. 2020 Aug;14(12):1099-1108. doi: 10.2217/bmm-2020-0205. PMID: 32969247.

The paper is interesting, but it has defects and before its possible publication, the Authors should respond to the following comments.

MAJOR:

1. There are a large number of abbreviations throughout the manuscript, and they are not always explained the first time they are used in the text. It would be a great help for a potential reader to introduce a list of abbreviations.

2. In the text of their manuscript, the authors use drug names, both international Nonproprietary Names (INNs) and trade names. For each of the listed drugs, the international name of the drug should be used and its trade name in brackets.

3. In the Introduction, the pathogenesis of secondary anemia in the course of CKD should be described in more detail, pointing to those etiological factors that are places of reference for pharmacological interventions, discussed later in the manuscript.

4. Chapter 2 - Overview of pharmacogenetics - is too general, different drug reactions also depend on age, gender, physiological and pathophysiological aspects.

5. Table 1 - no caption, The table is de facto a figure, not an editable text. Not all abbreviations are understandable - what is 5D (CKD stage), TSAT, ESA?

6. Figure 2 - no abbreviations (PH, HIF) in the description; what are the numbers in the selected fields (100, 101, 103, 104)? There are no such items in the reference list.

7. Figure 1 - the quality of the figure is insufficient, it should be prepared in a better resolution.

8. Table 2 - The authors inconsistently give the results - only for the study by Funakoshi et al. numerical efficacy values are given for the dependency on the blood group of patients, in other cases short conclusions are given in a descriptive form.

MINOR:

1. There are double spaces in the work - Authors should review the entire text for their removal.

2. Lines 105 – 109 – (…) This is because patients with CKD may have other underlying conditions that can affect the absorption and utilization of iron (…) – what factors affect iron absorption after IV administration?

3. Lines 118-119 (…) renal EPO-producing cells (REPs) are peritubular interstitial fibroblast-like cells and pericytes (…) - the localization of these cells in the kidneys should be clarified

4. Line 133 – (…) Pathophysiology of inflammation and linked genetic factors (…) – a sentence repeating the name of the next subsection?

Author Response

Please see the attachment. Please find the changes/amendment in blue highlight in the re-submitted manuscript

Reviewer 3 Report

Comments and Suggestions for Authors

Dear Authors, I have read with interest your manuscript. The paper addresses an very interesting issue regarding the treatment of anemia, considering it is both a consequence of the progression of chronic kidney disease but also a negative prognostic factor, with high mortality.

I would like to address a few suggestions/ questions:

·         The title is too complicated, I suggest to rethink the title.

·         As studies have shown, patients with chronic kidney disease have an extremely high genetic polymorphism, do you think that the patient with chronic kidney disease can also benefit from gene therapy?

·         Is gene therapy cost effective ?

·         Most studies with new molecules include patients with chronic kidney disease who do not require renal replacement therapy; when it comes to dialysis patients, anemia can contribute to increased cardiovascular risk – we also need to focus on this patients.

 This topic is very interesting to discus, and chronic anemia is a subject that should bring more specialists to the same table. I congratulate you on writing this paper.

Reviewer 4 Report

Comments and Suggestions for Authors

This is a nice paper. However, I have some comments. The findings from this paper are excellent and worthy to review. This manuscript contained some questions described below. I think this paper is interesting, this review contributes to future's clinical medicine largely. I have some questions from a point of view of clinical medicine. HIF-PH inhibitors are being looked at as a new treatment option for renal anemia. It is recognized as being of particular value in patients with ESA-resistant anemia and CKD with reduced iron utilization. On the other hand, attention should be paid to side effects, and the following point should be added in the description of HIF-PH inhibitors.

Please provide a discussion on the type of CKD-induced renal anemia for which they are highly useful. In particular, please add a note on the differences in therapeutic efficacy of HIF-PH inhibitors in CKD with high inflammatory conditions and CKD with reduced iron utilization, as well as in different HIF-PH subforms.2 With regard to the HIF-PH sub-forms, the action on the hypoxic response not only in the kidney, but also in the whole body, is a key factor in anemia Please add a note on the possibility of thromboembolic risk as opposed to improvement. Please also add a note on the reno-protective effect of HIF-PH inhibitors and the inhibition of interstitial fibrosis, which has been noted in some animal studies.

Round 2

Reviewer 1 Report

Comments and Suggestions for Authors

There is no response to the comment #5. Author should separate the table including clinical data and that including basic experiments.

Author Response

Basic experiments and clinical trial has been given separate titles to distinguish both

Reviewer 2 Report

Comments and Suggestions for Authors

Thank you for submitting the updated version of the manuscript. The authors took into account my suggestions. I believe that the paper in its present form meets the requirements for publication in MDPI.

Author Response

Dear reviewer, thank you very much for your feedback and recommendation.